# Distinct modes of endocytotic presynaptic membrane and protein uptake at the calyx of Held terminal of rats and mice

Yuji Okamoto[1], Noa Lipstein[2], Yunfeng Hua[3,4], Kun-Han Lin[4], Nils Brose[2], Takeshi Sakaba[1], Mitsuharu Midorikawa[1]*

[1]Graduate School of Brain Science, Doshisha University, Kyoto, Japan; [2]Department of Molecular Neurobiology, Max Planck Institute of Experimental Medicine, Göttingen, Germany; [3]Department of Connectomics, Max Planck Institute of Brain Research, Frankfurt, Germany; [4]Max-Planck-Institute for Biophysical Chemistry, Göttingen, Germany

**Abstract** Neurotransmitter is released at synapses by fusion of synaptic vesicles with the plasma membrane. To sustain synaptic transmission, compensatory retrieval of membranes and vesicular proteins is essential. We combined capacitance measurements and pH-imaging via pH-sensitive vesicular protein marker (anti-synaptotagmin2-cypHer5E), and compared the retrieval kinetics of membranes and vesicular proteins at the calyx of Held synapse. Membrane and Syt2 were retrieved with a similar time course when slow endocytosis was elicited. When fast endocytosis was elicited, Syt2 was still retrieved together with the membrane, but endocytosed organelle re-acidification was slowed down, which provides strong evidence for two distinct endocytotic pathways. Strikingly, CaM inhibitors or the inhibition of the $Ca^{2+}$-calmodulin-Munc13-1 signaling pathway only impaired the uptake of Syt2 while leaving membrane retrieval intact, indicating different recycling mechanisms for membranes and vesicle proteins. Our data identify a novel mechanism of stimulus- and $Ca^{2+}$-dependent regulation of coordinated endocytosis of synaptic membranes and vesicle proteins.

*For correspondence: mmidorik@mail.doshisha.ac.jp

**Competing interests:** The authors declare that no competing interests exist.

## Introduction

Synaptic transmission relies on neurotransmitter release from presynaptic terminals, which is mediated by exocytotic fusion of synaptic vesicles (SVs) with the plasma membrane. Following exocytosis, vesicular components are retrieved by endocytosis, and recycled for subsequent rounds of exocytosis (*Südhof, 2004*). SV recycling has been studied for decades using electron microscopy (*Heuser and Reese, 1973*; *Ceccarelli et al., 1973*; *Takei et al., 1998*; *Schikorski and Stevens, 2001*; *Watanabe et al., 2013*), membrane capacitance measurements (*von Gersdorff and Matthews, 1994*; *Moser and Beutner, 2000*; *Sun and Wu, 2001*), and live imaging of fluorescent markers (*Betz and Bewick, 1992*; *Miesenböck et al., 1998*; *Sankaranarayanan and Ryan, 2001*; *Hua et al., 2010*).

Two major modes of vesicle fusion have been proposed, kiss-and-run, in which the membrane fusion occurs transiently and vesicle membrane is retrieved simply by fusion pore closure (*Fesce et al., 1994*), and full fusion, where the vesicle membrane collapses into the plasma membrane (*Cremona and De Camilli, 1997*). In the kiss-and-run scenario, the fusing vesicles are thought to largely maintain their identity as regards membrane and protein composition, but it is not clear how membrane and protein retrieval is coordinated when full fusion occurs.

**eLife digest** Nerve cells release chemicals called neurotransmitters to communicate with each other. The neurotransmitters are packaged inside membrane-encased sacs called vesicles that fuse with the cell's membrane and release their contents into the space between the nerve cells. The vesicle membrane (which also has proteins embedded in it) can then be retrieved into the cell, and recycled to make new vesicles ready to release more neurotransmitters.

Recycling vesicle components requires highly coordinated mechanisms that regulate how much membrane and vesicle protein is retrieved from the cell membrane. Researchers interested in these processes have often studied them at a part of the brainstem of mammals known as the calyx of Held. However, many of the details about how vesicle proteins are recycled remained unclear.

Okamoto et al. have now measured vesicle membrane and protein retrieval at the same time and in the same cell at the calyx of Held from rats and mice. The cell surface area was also measured, and the experiments focused on a fluorescently tagged version of a vesicle protein called Synaptotagmin2 that could be tracked under a microscope. Okamoto et al. found that, in weakly active nerve cells, the vesicle membrane and Synaptotagmin2 were retrieved together at a slow rate. The process was faster in more active nerve cells, and Synaptotagmin2 was still retrieved with the membrane but it appeared to be stored first in larger sacs. This suggests that membrane and vesicle proteins may be retrieved via two distinct modes depending on the activity strength. The results of further experiments went on to suggest that vesicle membranes might be recycled in a different way from vesicle proteins.

Finally, Okamoto et al. also found a signaling pathway that couples the uptake of vesicle membrane with uptake of Synaptotagmin2. Future studies could now explore how these processes work in other types of nerve cell.

As for membrane retrieval, capacitance measurements revealed a fast and a slow endocytosis component. Fast endocytosis with a time constant of less than few seconds, is seen when a strong stimulus is applied (*Xue et al., 2012*), and often involves excess membrane retrieval. In contrast, slow endocytosis, which is a major form of endocytosis, has a time constant of seconds to tens of seconds, and at a variety of synapses, the amount of membrane endocytosed in this manner is similar to the amount of membrane that had previously been exocytosed (*von Gersdorff and Matthews, 1994*; *Moser and Beutner, 2000*; *Sun and Wu, 2001*). Subsequent studies showed that fast and slow endocytosis reflects clathrin-independent and clathrin-dependent modes of endocytosis, respectively (*Wu et al., 2009*; *Yamashita et al., 2010*), and that the contribution of fast endocytosis increased progressively by applying stronger stimulation (*Renden and von Gersdorff, 2007*; *Wu et al., 2009*; *Yamashita et al., 2010*; *Midorikawa et al., 2014*). Precise control of the amount of membrane retrieval has high physiological relevance, since perturbation of membrane retrieval induces use-dependent depletion of the releasable vesicles and rundown of exocytosis (*Yamashita et al., 2005*).

The maintenance of transmitter release is not only dependent on membrane retrieval following membrane fusion, but also on the recovery of vesicular proteins into the recycling vesicles (*Kononenko and Haucke, 2015*). Vesicle protein recycling has been studied mainly by using a pH-sensitive green fluorescent protein (GFP) variant, pHluorin, as a vesicle protein marker (*Miesenböck et al., 1998*). Except for kiss-and-run events, retrieval of vesicular proteins seems to be slow, with a time constant of seconds to tens of seconds, similar to the time constant of membrane retrieval during slow endocytosis as assessed by capacitance measurements. However, in contrast to membrane retrieval, the retrieval of synaptic proteins slows down as stimulation persists (*Armbruster et al., 2013*; *Fernández-Alfonso and Ryan, 2004*). Further, exocytosed vesicular proteins, such as VAMP2 (*Sankaranarayanan and Ryan, 2001*; *Gandhi and Stevens, 2003*), synaptophysin (*Granseth et al., 2006*), synaptotagmin (*Fernández-Alfonso et al., 2006*), and VGLUT (*Balaji and Ryan, 2007*), appear to be retrieved into endocytotic organelles to similar extents. Proper endocytotic retrieval is of high physiological relevance, given that clearance of vesicle proteins from transmitter release sites is required for the maintenance of synaptic transmission and that

insufficient retrieval leads to the slowed recruitment of SVs to release sites (*Hosoi et al., 2009*; *Wu et al., 2009*; *Hua et al., 2013*).

Previous studies on endocytotic membrane retrieval based on capacitance measurements and on endocytotic protein recycling indicate the presence of highly coordinated mechanisms that regulate the extent of membrane and protein retrieval after exocytosis. Several endocytotic proteins, such as AP-2 (*Traub, 2009*), AP180 (*Koo et al., 2011*), endophilin (*Milosevic et al., 2011*), or stonin 2 (*Kononenko et al., 2013*), were proposed as potential adaptors for endocytocic vesicle protein sorting, but the detailed mechanisms are still debated (*Opazo and Rizzoli, 2010*). In addition, little is known about corresponding upstream modulatory or regulatory processes and their activity dependence.

To gain detailed insight into the mechanisms of exo-endocytosis coupling, simultaneous measurements of membrane retrieval and vesicle protein retrieval are required that allow for the separate assessment of membrane retrieval kinetics and vesicle protein retrieval kinetics in the same cell. Such simultaneous measurements would further allow us to address the known kinetic differences between capacitance measurements and pHluorin-based measurements.

The calyx of Held synapse is a mammalian model system of synaptic transmission at central synapses (*Forsythe, 1994*; *Borst and Sakmann, 1996*; *Wu and Borst, 1999*). The large terminal size (10–20 µm) enables simultaneous voltage-clamp recordings from pre- and post-synaptic compartments, so that the kinetics of exocytosis can be dissected (*Wu and Borst, 1999*; *Sakaba and Neher, 2001*). In addition, the extent and time course of membrane fusion and retrieval can be monitored by membrane capacitance measurements (*Sun and Wu, 2001*). Correspondingly, the calyx of Held is one of the best characterized synapses as regards the relationship between the kinetics of exo- and endocytosis. However, the dynamics of endocytotic vesicle protein recycling has remained unclear even in this synapse.

To monitor the dynamics of endocytotic vesicle protein recycling at the calyx terminal, we used the cypHer5E fluorophore (referred hereafter as cypHer; *Adie et al., 2003*; *Hua et al., 2011*), as an exo-endocytosis reporter. The cypHer moiety, which has a pH dependence opposite to pHluorin (i.e. fluorescent in acidic pH, and quenched in neutral pH), was coupled to antibodies against the luminal domain of Synaptotagmin 2 (anti-Syt2-cypHer), an endogenous SV protein at the calyx of Held terminal (*Pang et al., 2006*), so that our strategy directly labels an endogenous vesicle protein with cypHer. We combined capacitance measurements and pH-sensitive vesicle protein imaging at the calyx of Held presynaptic terminal to simultaneously monitor the kinetics of membrane and vesicular protein exo-endocytosis. Upon stimulation, we found a 12–14 s delay followed by a 20–40 s decay time constant for re-acidification of retrieved Syt2-containing organelle. By comparing the time courses of endocytotic membrane and Syt2 uptake, we discovered distinct modes of membrane and protein uptake depending on stimulus intensities, and key signaling mechanisms that maintain the coordinated uptake.

## Results

### Simultaneous recordings of membrane capacitance and anti-Syt2-cypHer uptake

To visualize the turnover of the vesicular protein Syt2, we labeled calyx terminals with anti-Syt2-cypHer in a slice preparation (*Figure 1A*). The 200 µm transverse brainstem slices were incubated for 30 min in a high potassium solution (32.5 mM) containing anti-Syt2-cypHer (0.01 mg/ml) to depolarize the terminals and induce exocytosis followed by endocytosis, in which the anti-Syt2-cypHer is internalized (*Figure 1B*). The fluorescence of the cypHer dye is almost quenched at the neutral extracellular pH of 7.4 after SV exocytosis and is almost maximal at the intravesicular pH of 5.5 (*Hua et al., 2011*). Therefore, the cup-shaped structures observed in *Figure 1A* can be assumed to be calyx terminals filled with anti-Syt2-cypHer-containing internalized vesicles. The staining was calyx specific in this region, and anti-Syt2-cypHer showed a stimulus dependent fluorescence change when field stimulations were applied (*Figure 1—figure supplement 1*; *Hua et al., 2011*). Anti-Syt2-cypHer exposed to the membrane surface is barely fluorescent, so that background fluorescence is kept low even in a slice preparation. To measure the kinetics of Syt2 and plasma membrane recycling simultaneously, labeled calyx terminals were whole-cell voltage clamped, and the membrane

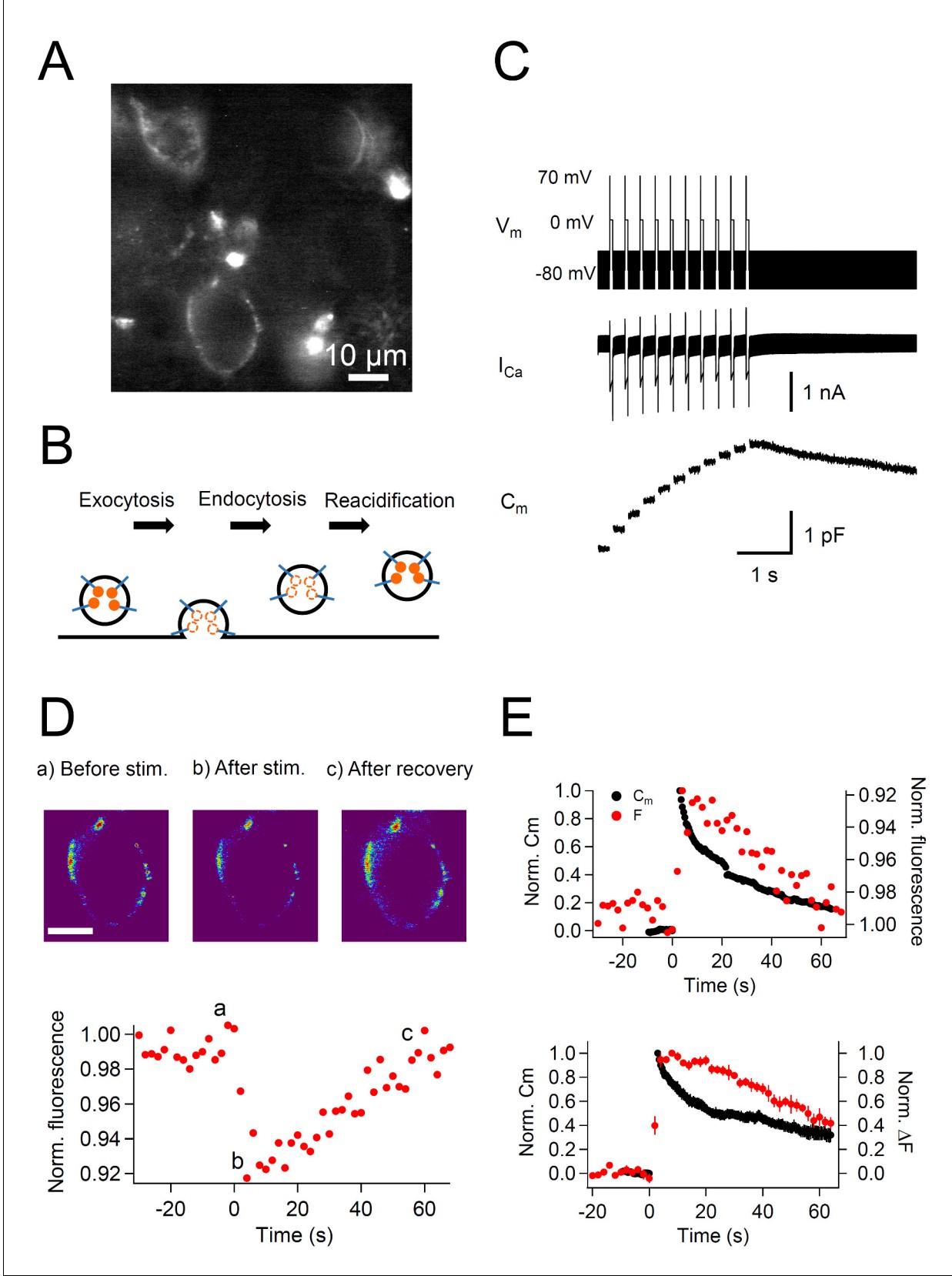

**Figure 1.** Simultaneous recording of membrane capacitance and anti-Syt2-cypHer fluorescence. (**A**) Example fluorescence image of calyx of Held presynaptic terminals labeled with anti-Syt2-cypHer. Intravesicular anti-Syt2-cypHer emits fluorescence upon excitation at 645 nm.(**B**) Schematic view of
*Figure 1 continued on next page*

*Figure 1 continued*

fluorescence changes of anti-Syt2-cypHer during exo-endocytosis. The orange dots show cypHer coupled to antibodies against the luminal domain of Syt2. The cypHer fluorescence is maximal at intravesicular pH 5.5 and almost quenched at the extracellular pH 7.4. Upon exocytosis, the fluorescence is quenched because of the exposure to the extracellular pH. During endocytosis and re-acidification, the fluorescence is de-quenched again. (C) A train of depolarizing pulses (0 mV for 50 ms following a prepulse to +70 mV for 2 ms, 10 stimuli, interstimulus interval 200 ms, $V_m$) was applied to elicit a $Ca^{2+}$ current ($I_{Ca}$), and membrane capacitance ($C_m$) was measured during the sweep. The prepulse (+70 mV) was applied to activate $Ca^{2+}$ channels maximally without causing $Ca^{2+}$ influx. A sine wave (30 mV in amplitude, 1,000 Hz in frequency) was superimposed on a holding potential of -80 mV to measure membrane capacitance ($C_m$). (D) The top panel shows example fluorescence images showing the cypHer fluorescence image (a) before stimulation, (b) after stimulation, and (c) after recovery, shown in a pseudo-colored scale. Each image was taken at the time point shown in the bottom trace. Scale bar, 10 μm. The bottom panel shows an example of a normalized fluorescence trace of anti-Syt2-cypHer in response to a train of depolarizing pulses. The fluorescence intensity was normalized to the first point in the plot. (E) The top panel shows example traces of normalized $C_m$ (black circles, left axis) and cypHer fluorescence (red circles, right axis) at a calyx terminal stimulated by a train of depolarizing pulses. The $C_m$ trace was normalized to the amplitude of the capacitance jump, and the fluorecence trace was normalized to the initial intensity. The fluorescence trace was inverted to compare the time courses of $C_m$ and fluorescence traces.The bottom panel shows average traces of normalized $C_m$ (black circles, n = 7) and cypHer fluorescence change (red circles, n = 10) at the calyx terminal evoked by a train of depolarizing pulses (7 data were obtained from simultaneous measurements of capacitance and cypHer). $C_m$ traces were normalized to the peak capacitance change (left axis), and fluorecence traces were normalized to the peak fluorescence change (right axis).

The following figure supplements are available for figure 1:

**Figure supplement 1.** Calyx specific anti-Syt2-cypHer staining and fluorescence change evoked by field stimulation.

**Figure supplement 2.** Comparison of the recovery time course of capacitance and cypHer (rat).

**Figure supplement 3.** CypHer signal recovery after a train of depolarizing pulses (longer recording).

capacitance and the cypHer signal were measured. When the terminal was stimulated with a train (10 times, 200 ms intervals) of depolarization pulses (0 mV for 50 ms following a prepulse to +70 mV for 2 ms), large capacitance jumps caused by exocytosis were observed (1.52 ± 0. 21 pF after 10 pulses, n = 7, *Figure 1C*). A 50 ms pulse is sufficient to deplete the readily-releasable pool of SVs (*Hosoi et al., 2009*; *Wu et al., 2009*; *Yamashita et al., 2010*), and the following pulses mainly reflect release of newly-replenished SVs at release sites. The capacitance decayed bi-exponentially with two time constants of a few seconds (τ = 5.2 s, 37% ) and tens of seconds (τ = 66.5 s, 63% ), reflecting clathrin-independent and dependent endocytosis, respectively (*Wu et al., 2009*; *Yamashita et al., 2010*). During the recording, we also measured the fluorescence of the cypHer signal. The cypHer fluorescence showed a rapid decrease upon stimulation, followed by a slow recovery (*Figure 1D*). The time course of fluorescence change reflects the kinetics of exocytosis and subsequent endocytosis. To compare the time course of membrane capacitance traces and cypHer fluorescence traces, we inverted the cypHer traces. Comparison of the recovery time course of the capacitance traces and the cypHer traces showed that both returned to ∼0.4 of the peak value after 60 s ($C_m$, 0.33 ± 0.06, n = 7; F, 0.47 ± 0.07, n = 10, *Figure 1E*). Two notable features are revealed by comparing capacitance and cypHer measurements: (1) $C_m$ had a rapid decay component whereas cypHer signals did not (see also *Figure 4*. below); (2) CypHer signals showed a delay in the decay phase as compared with the decay of the capacitance values. The delayed recovery of cypHer signal is consistent with reports showing that acidification of glutamatergic SVs occurs with a time constant of tens of seconds (*Egashira et al., 2015*, but see *Atluri and Ryan, 2006*). Fitting of the recovery time course of the cypHer signal was optimal when we assumed a re-acidification time constant of 38.7 s and a 14 s delay of onset after membrane retrieval (*Figure 1—figure supplement 2*). The asymptotic value (extrapolated from the exponential fit) of the fluorescence recovery (0.13 from *Figure 1—figure supplement 3*) was similar to that of the capacitance trace (0.12 from *Figure 1E*), which indicated that the rate of endocytosis becomes as low as that of re-acidification. The exponential decay time constant was more clearly seen by taking longer recordings (*Figure 1—figure supplement 3*), suggesting that exocytosed Syt2 is almost completely retrieved.

To investigate the time course of Syt2 recycling upon exo-endocytosis using anti-Syt2-cypHer, we have to assume that anti-Syt2-cypHer labeling of the vesicles is homogenous. To verify this, we compared the magnitude of the capacitance jumps with the amplitude of cypHer fluorescence to see

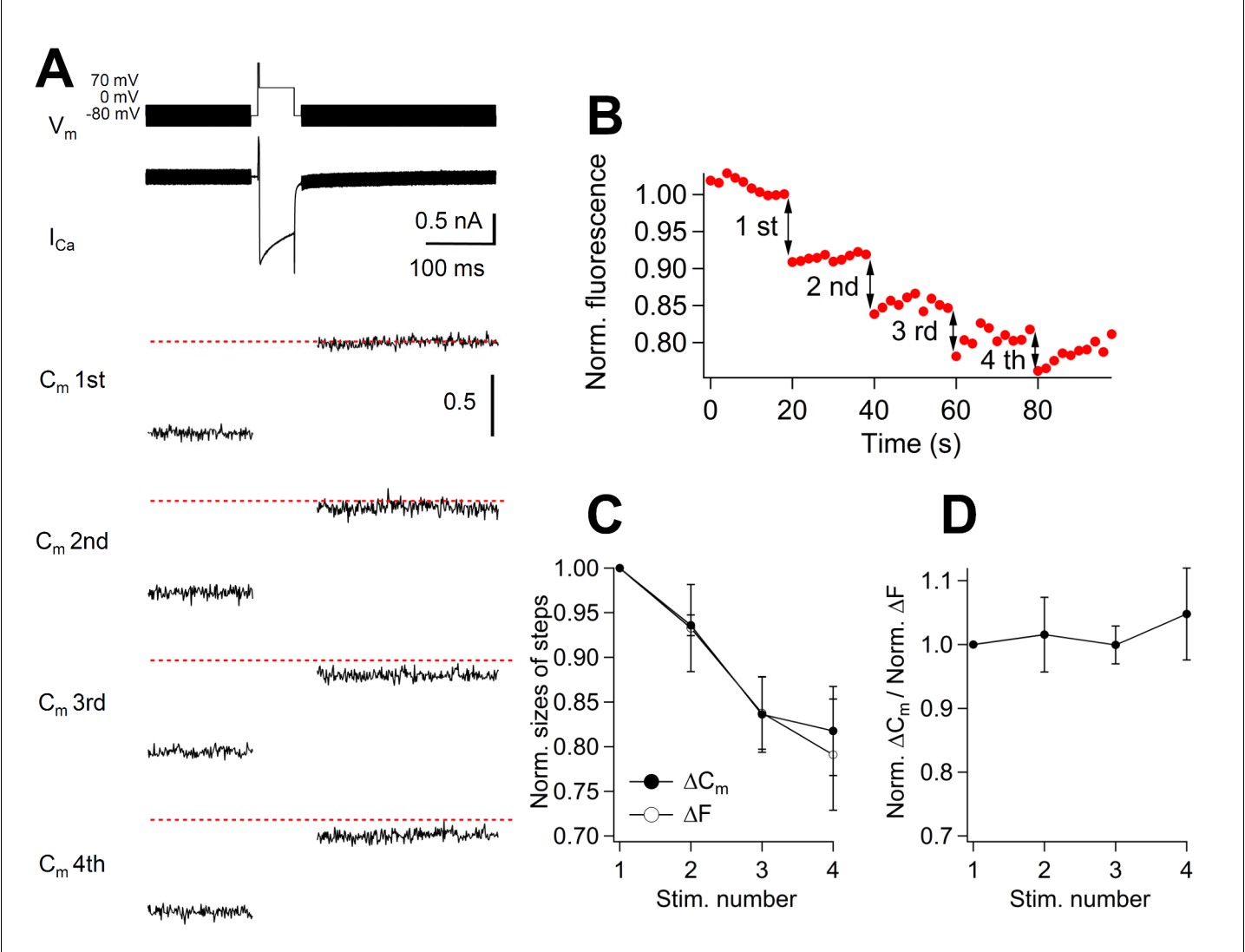

**Figure 2.** Correlation between the amount of membrane capacitance changes and cypHer signal changes. (**A**) A 50 ms depolarizing pulse (0 mV for 50 ms, following a prepulse to +70 mV for 2 ms), as shown on the top, was applied four times with an inter stimulus interval of 20 s. The bottom panels show averaged capacitance traces of four consecutive stimulations (n = 6). Traces from each cell were normalized to the first $\Delta C_m$ amplitude. The dotted red lines show the first $\Delta C_m$ amplitudes. (**B**) Example of cypHer fluorescence changes induced by four consecutive 50 ms depolarizations. (**C**) The extent of capacitance jumps (filled circles) and cypHer fluorescence changes (open circles) evoked by four consecutive depolarizations. (**D**) The ratio of normalized $\Delta C_m$ and normalized $\Delta F$ of the cypHer signals plotted against the stimulus number.

whether they correlate linearly. We applied four consecutive 50 ms depolarizing pulses with 20 s intervals, and simultaneously measured the size of the capacitance jumps and the cypHer fluorescence change. With this protocol, capacitance jumps showed a gradual decrease because of incomplete recovery of the readily releasable vesicle pool between the stimuli (*Figure 2A*). The amplitudes of the cypHer signal also showed gradual decrease (*Figure 2B*). When the normalized amplitudes of capacitance jumps and fluorescence decrease were compared, we found similar levels of depression (*Figure 2C,D*). This result indicates that the membrane capacitance changes and the changes in cypHer fluorescence amplitudes correlate linearly, even under conditions of a slight change in the extent of exocytosis ($\sim 20\%$), and that the cypHer labeling of the vesicles is homogenous.

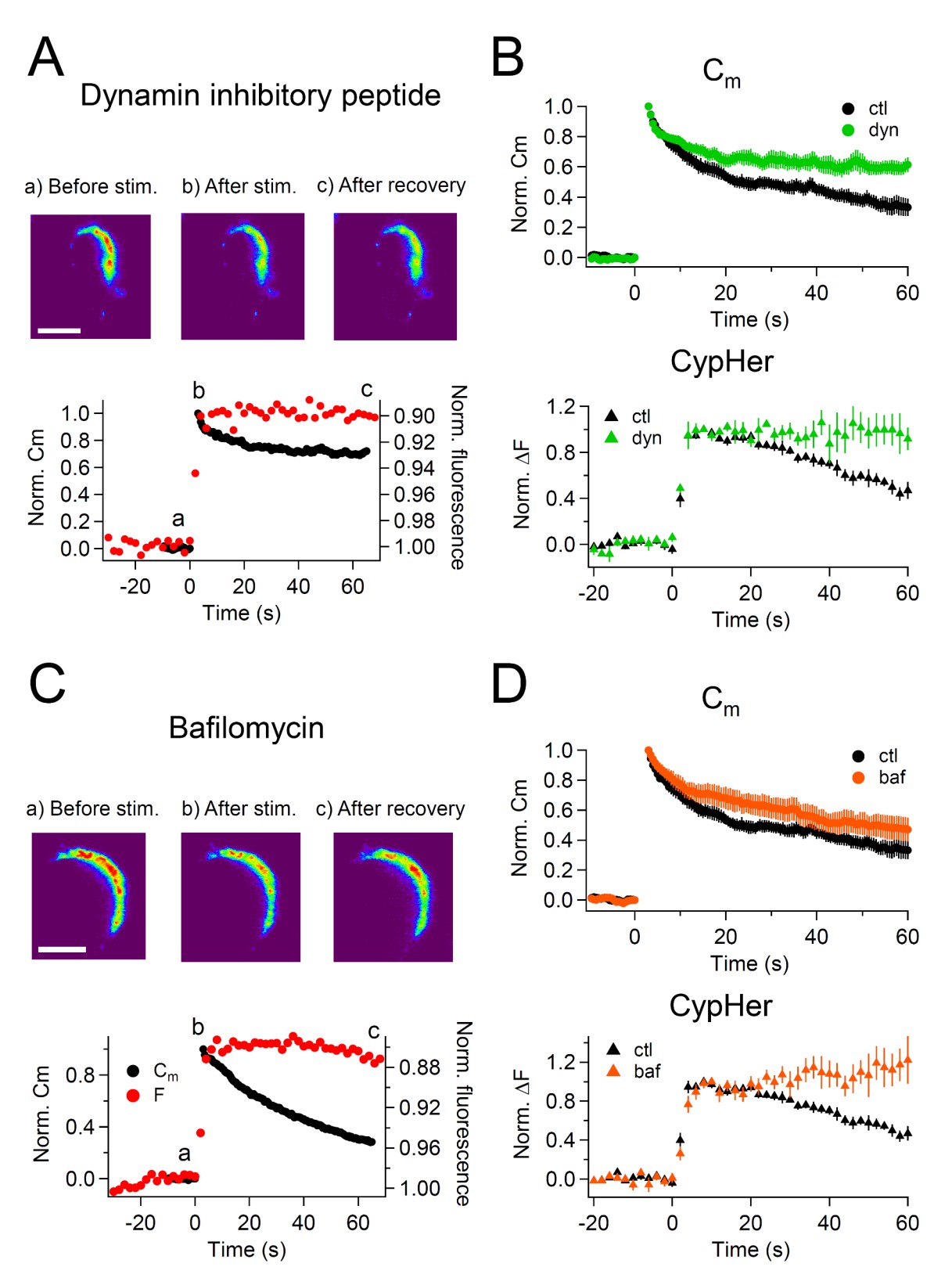

**Figure 3.** CypHer signal recovery depends on endocytosis and re-acidification. (**A**) A train of depolarizing pulses (see *Figure 1C*) was applied to elicit exocytosis in the presence of 1 mM dynamin inhibitory peptide. Images in the top panel show the cypHer fluorescence at the time point indicated in
*Figure 3 continued on next page*

*Figure 3 continued*

the plot below (see also *Figure 1B*). Scale bar, 10 μm. The bottom panel shows example traces of $C_m$ and cypHer fluorescence in the presence of dynamin inhibitory peptide. (B) The top panel shows averaged $C_m$ traces under control conditions (n = 7, black circles) and in the presence of dynamin inhibitory peptide (n = 4, green circles). The bottom panel shows cypHer fluorescence changes under control conditions (n = 10, black triangles) and in the presence of dynamin inhibitory peptide (n = 4, green triangles). (C) Same as in A, with 5 μM bafilomycin applied during the recording. (D) The top panel shows averaged $C_m$ traces under control conditions (n = 7, black circles) and in the presence of bafilomycin (n = 5, orange circles). The bottom panel shows averaged fluorescence traces under control conditions (n = 10, black triangles) and in the presence of bafilomycin (n = 5, orange triangles).

## The effects of blocking endocytosis and re-acidification on cypHer signals

Next, we examined if the recovery of the cypHer signal is caused by endocytosis and subsequent re-acidification. For this purpose, we blocked endocytosis and re-acidification by using pharmacological tools.

First, we inhibited endocytosis by applying dynamin inhibitory peptide (1 mM) intracellularly through the patch pipette. Dynamin is essential for slow endocytosis at the calyx terminal (*Yamashita et al., 2005*; *2010*; *Wu et al., 2009*). After a 4–5 min waiting period to allow the peptide to diffuse throughout the terminal, we applied a train of ten 50 ms depolarizations. Dynamin inhibitory peptide inhibited endocytosis following a capacitance jump except for the early fast component (*Figure 3A*), which is consistent with previous reports (*Yamashita et al., 2010*), and the cypHer signal recovery was also blocked by dynamin inhibitory peptide (*Figure 3A*). Both the capacitance and the cypHer signal recovery were blocked, and their recovery became significantly smaller in the presence of dynamin inhibitory peptide than under control conditions (*Figure 3B*). Normalized $C_m$ after 60 s was 0.33 ± 0.06 for control calyces (n = 7), and 0.61 ± 0.05 in calyces treated with dynamin inhibitory peptide (n = 4, p<0.01), while the cypHer signal recovered to a value of 0.47 ± 0.07 in control (n = 10) and to 0.92 ± 0.10 in the presence of dynamin inhibitory peptide (n = 4, p<0.01).

Next, we blocked the re-acidification of endocytosed vesicles by bath application of bafilomycin, a V-type ATPase inhibitor (5 μM). We applied bafilomycin for 4–5 min before the recordings to block re-acidification of newly endocytosed vesicles. Upon application of a train of ten 50 ms depolarizations, capacitance showed a robust jump and a clear recovery after the jump, indicating that membrane retrieval was still functional in the presence of bafilomycin (*Figure 3C*). On the other hand, the cypHer signal did not recover, as expected under blockade of re-acidification by bafilomycin (*Figure 3C*; *Sankaranarayanan and Ryan, 2001*). Average traces showed no difference in the time course of capacitance recovery (normalized $C_m$ after 60 s; control, 0.33 ± 0.06, n = 7; bafilomycin, 0.47 ± 0.08, n = 5, p=0.19), but the recovery of cypHer signal was blocked with bafilomycin (normalized ΔF after 60 s; control, 0.47 ± 0.10, n = 10; bafilomycin, 1.22 ± 0.24, n = 5, p=0.03).

Based on these results, we conclude that the cypHer signal recovery results from endocytosis and subsequent re-acidification of the endocytosed organelles.

## Syt2 is taken up into slowly re-acidifying organelles after prolonged depolarization

Because the fast and the slow components endocytosis were co-detected with the capacitance measurements (*Figure 1E*), while the fast mode was not seen in the cypHer measurements (*Figure 3B*), we next examined more specifically the retrieval of Syt2 during fast endocytosis.

The fast mode of endocytosis occurs predominantly in response to strong stimulation in P8-11 calyx terminals, and is thought to be triggered by elevation of the bulk cytoplasmic $Ca^{2+}$ concentration (*Wu et al., 2005*). This mode of endocytosis retrieves the plasma membrane extremely fast, which can be read out as a rapid membrane capacitance decay after the exocytotic capacitance jump, and often shows an undershoot (*Renden and von Gersdorff, 2007*; *Wu et al., 2009*). It has been shown that the amount of fast mode of endocytosis was gradually increased by elongating pulse durations in P8-11 calyx of Held terminals (*Renden and von Gersdorff, 2007*; *Wu et al., 2009*; *Yamashita et al., 2010*; *Midorikawa et al., 2014*). While the kinetics of this membrane retrieval has been characterized, the kinetics of the accompanying protein retrieval is largely unknown.

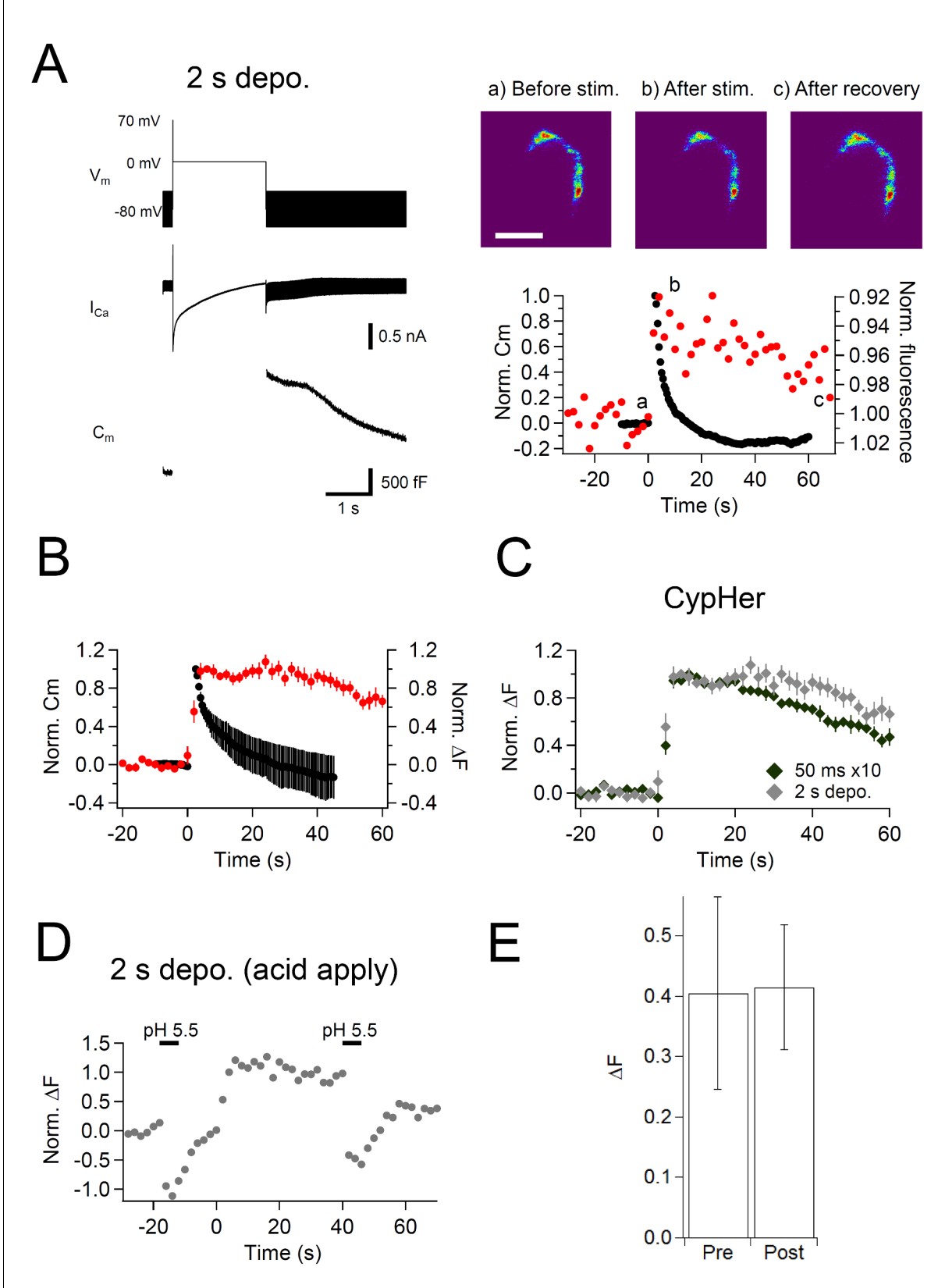

**Figure 4.** Effect of a prolonged depolarizing pulse on the kinetics of capacitance and cypHer signal recovery. (**A**) Similar to *Figure 1*, but a single prolonged (2 s) depolarization ($V_m$) was applied, which induced large calcium currents ($I_{Ca}$) and capacitance jumps ($C_m$). Images (right top) show the

*Figure 4 continued on next page*

*Figure 4 continued*

cypHer fluorescence at each time point shown in the panel below. Scale bar, 10 µm. The right bottom panel shows example traces of $C_m$ and cypHer fluorescence obtained with the 2 s depolarization paradigm. (**B**) Average traces of $C_m$ (black circles) and cypHer fluorescence (red circles) at the calyx terminal evoked by a 2 s depolarizing pulse (n=7). (**C**) Averaged cypHer fluorescence traces obtained by applying a train of depolarizing pulses (the same as *Figure 1C,* n=10, black diamonds) or by a 2 s depolarizing pulse (n=7, gray diamonds). CypHer fluorescence recovery was slower after a 2 s depolarization (40 s after the stimulus; 50 ms x 10, 0.70 ± 0.04; 2 s, 0.93 ± 0.07, p=0.019). (**D**) Example trace of cypHer fluorescence change when an acidic extracellular solution (pH 5.5) was puff-applied during the recording. The pH 5.5 solution was puff-applied twice for 5 s, 20 s before and 40 s after stimulation. (**E**) Comparison of ΔF induced by the puff-application of pH 5.5 solution before (Pre) and after (Post) stimulation. There was no significant difference (Pre, 0.40 ± 0.16; Post, 0.41 ± 0.10, n = 5, p=0.66).

The following figure supplement is available for figure 4:

**Figure supplement 1.** CypHer signal recovery after a prolonged depolarizing pulse (longer recording).

In this set of experiments, the calyx terminal was stimulated by a 2 s depolarization pulse to induce only a fast mode of endocytosis (*Yao and Sakaba, 2012*). The membrane capacitance showed a large jump upon the stimulation (>1 pF, *Figure 4A*, left), and decayed rapidly after the stimulation, which is consistent with previous studies. This fast mode of endocytosis is shown to be insensitive to dynamin inhibition (*Yamashita et al., 2010*, but see *Xu et al., 2008*), as in *Figure 3* fast component, suggesting the same mechanism. On the contrary, the cypHer signal barely showed a recovery (*Figure 4A*, right). The discrepancy was more clearly seen when we compared the averaged traces of membrane capacitance and cypHer signal (*Figure 4B*). This is different from the results obtained with trains of ten 50 ms depolarizations (*Figure 1*), where membrane capacitance and cypHer signal decayed with more similar time courses. In fact, the recovery time course of cypHer signal after the 2 s depolarization was even slower than after trains of ten 50 ms depolarizations (*Figure 4C*, 40 s after the stimulus; 50 ms x 10, 0.70 ± 0.04, n = 10; 2 s, 0.93 ± 0.07, n = 9, p=0.019). To explain the delayed cypHer signal recovery after the 2 s depolarization condition, we considered two possibilities: (i) the anti-Syt2-cypHer was not retrieved together with the endocytosed membrane and left behind at the plasma membrane surface, or (ii) the anti-Syt2-cypHer was retrieved via endocytosed organelles, but the intra-organelle pH was not (or barely) re-acidified after endocytosis. To discriminate between these two possibilities, we briefly (5 s) applied an acidic solution (pH 5.5) extracellularly before (20 s) and after (40 s) the stimulation to de-quench the surface-exposed anti-Syt2-cypHer. In the 2 s pulse condition, membrane retrieval assessed by capacitance measurements is essentially completed 40 s after the stimulation (*Figure 4B*). If the anti-Syt2-cypHer were taken up into the endocytosed organelle, there should be no difference in the amount of de-quenching before and after the stimulation. On the other hand, the amount of de-quenching should be higher after stimulation if anti-Syt2-cypHer were not taken up and left behind at the plasma membrane. We found (*Figure 4D and E*) that the amount of de-quenching was not different before or after the stimulation (pre, 0.40 ± 0.16; post, 0.41 ± 0.10, n = 5, p=0.66), indicating that anti-Syt2-cypHer was retrieved via endocytosed organelles, but was not (or barely) re-acidified. The lack of re-acidification in the newly endocytosed organelles after a 2 s depolarization can be explained if the membrane endocytosis occurred as bulk endocytosis, which is seen after strong stimulation of the calyx terminal (*de Lange et al., 2003*). Longer post-stimulus recordings showed a sign of cypHer signal recovery (*Figure 4—figure supplement 1*), which is consistent with the idea that anti-Syt2-cypHer is initially taken up via larger endosomal-like structure formed by bulk endocytosis, followed by slow re-acidification due to the larger volume-to-surface ratio.

## Effect of calmodulin blockade on Syt2 uptake

Calmodulin (CaM) plays a role during endocytosis, and its contribution is dependent on the age of the animals (*Yamashita et al., 2010*; *Yao and Sakaba, 2012*; but see *Wu et al., 2009*). In addition to its role in endocytosis itself, it was suggested that CaM is crucial for clearing fused vesicle membrane and proteins from release sites (*Wu et al., 2009*; *Hosoi et al., 2009*). To test the role of CaM in coordinated membrane and Syt2 retrieval during the slow mode of endocytosis, we inhibited CaM function by intracellular application of a CaM inhibitory peptide (20 µM). We whole-cell voltage clamped the calyx terminal and waited for 4–5 min to allow the peptide to diffuse into the terminal.

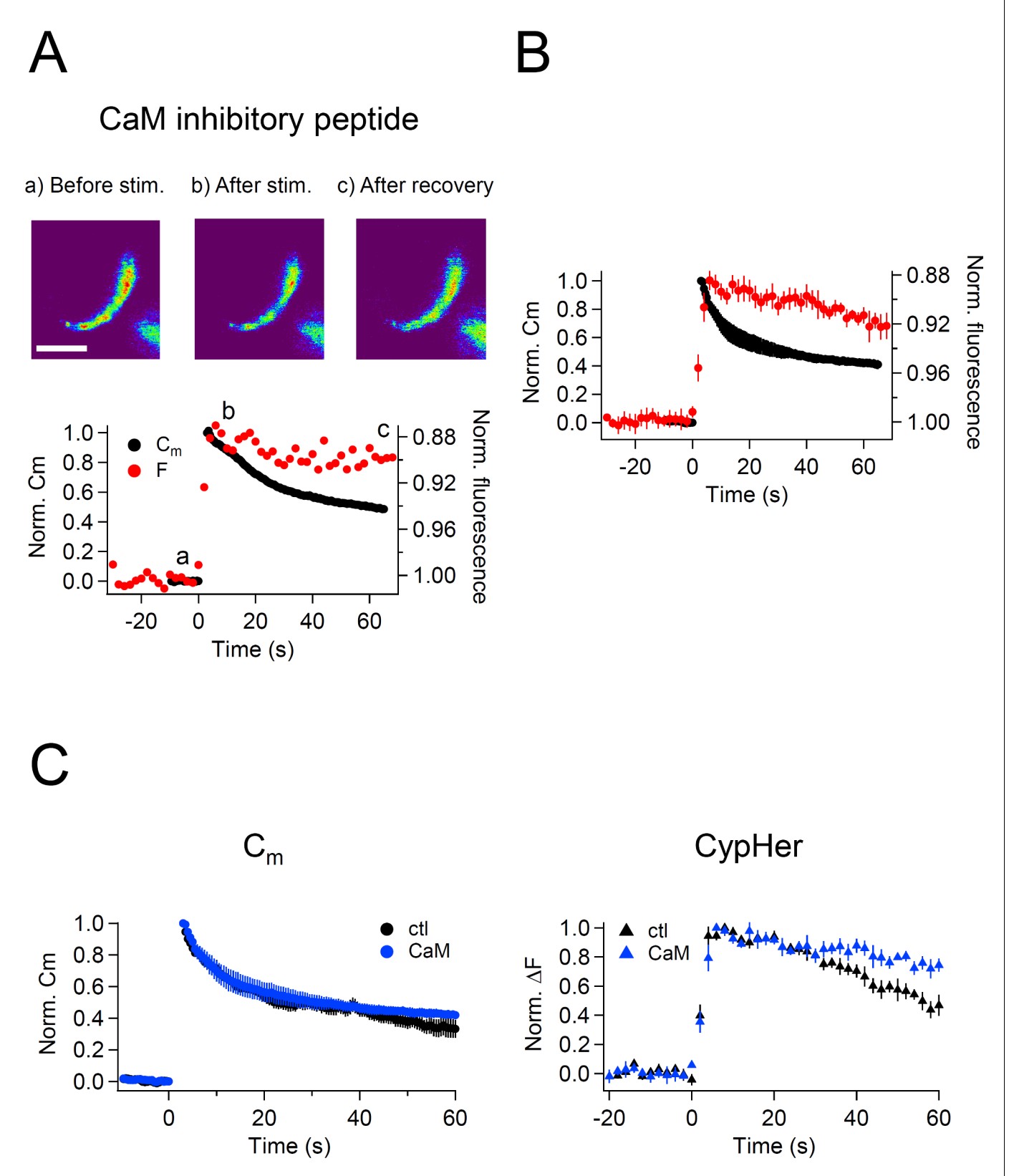

**Figure 5.** Effect of calmodulin inhibitory peptide on the kinetics of capacitance and cypHer signal recovery. (**A**) A train of depolarizing pulses (the same as *Figure 1C*) was applied to elicit exocytosis in the presence of 20 μM CaM inhibitory peptide. Images (top) show the cypHer fluorescence at each

*Figure 5 continued*

time point depicted in the bottom panel. The bottom panel shows example traces of C$_m$ and cypHer fluorescence in the presence of the CaM inhibitory peptide. Scale bar, 10 µm.(**B**) Average traces of C$_m$ (black circles) and cypHer fluorescence (red circles) in the presence of CaM inhibitory peptide (n = 5).C. The left panel shows averaged C$_m$ traces under control conditions (n = 7, black circles) and in the presence of CaM inhibitory peptide (n = 5, blue circles). The right panel shows averaged cypHer fluorescence traces under control conditions (n = 10, black triangles) and in the presence of CaM inhibitory peptide (n = 5, blue triangles).

Then, a train of ten 50 ms depolarizations was applied to evoke exocytosis followed by endocytosis, and the effects on membrane capacitance and cypHer signal recovery were monitored.

After the stimulus, membrane capacitance recovered, but the cypHer signal barely showed any recovery (*Figure 5A,B*). Averaged traces show that the time course of membrane retrieval was not affected (*Figure 5C* left, 60 s after the stimulus; control, 0.33 ± 0.06, n = 7; CaM inhibitory peptide, 0.42 ± 0.03, n = 5, p=0.21), but the cypHer signal recovery was reduced and/or slowed down by application of CaM inhibitory peptide (*Figure 5C* right, 60 s after the stimulus; control, 0.47 ± 0.07, n = 10; CaM inhibitory peptide, 0.74 ± 0.04, n = 5, p<0.01). It remains possible that a higher concentration of CaM inhibitor is required to inhibit membrane retrieval itself (*Sun et al., 2010*; *Wu et al., 2014*). To test if the slower recovery of the cypHer signal was caused by slower protein retrieval or else by slower re-acidification, we performed an acidic solution (pH 5.5) perfusion experiment as in *Figure 4D*. We applied acidic solution before (20 s) and after (40 s) the train of ten 50 ms depolarizations under control conditions in the absence (*Figure 6A*) or presence of CaM inhibitory peptide (*Figure 6B*). The amounts of de-quenching induced by acid perfusions, before and after stimulation, were measured in both conditions (*Figure 6A,B*). Then, we calculated the de-quenching ratio after and before the stimulation (post ΔF / pre ΔF), and compared the values between the two conditions. We found that the post ΔF / pre ΔF value was larger in the presence of CaM inhibitory peptide (control, 1.15 ± 0.20, n = 5; CaM inhibitory peptide, 1.95 ± 0.14, n = 5, p=0.012, *Figure 6C*). The small increase in the post ΔF / pre ΔF ratio under control condition is caused by incomplete membrane retrieval 40 s after the stimulus (*Figure 1*). Because the time course of membrane retrieval was similar for both control and CaM inhibitory peptide conditions (*Figure 5C*), this result indicates that anti-Syt2-cypHer was left behind at the plasma membrane surface in the presence of CaM inhibitory peptide. Thus, CaM may play a crucial role in coordinated retrieval of Syt2 together with the endocytosed membrane.

We also calculated the relative density of 'stranded' Syt2 present at the surface membrane before stimulation and that of the vesicular Syt2, based on the control condition data. The basal capacitance of the calyx terminal before the stimulation was 18.87 ± 1.56 pF, and the exocytotic capacitance jump caused by stimulation was 1.52 ± 0. 21 pF. When fluorescent change was normalized to the amount of quenching upon stimulation, the amount of de-quenching by acid perfusion before the stimulus was 0.48 ± 0.23. From these values, we calculated the amount of fluorescence per area. The normalized ΔF/pF was 0.026 ± 0. 012 for the surface stranded Syt2, and 0.71 ± 0.09 for vesicular Syt2. Thus, the density of vesicular Syt2 is ~30 times higher than the stranded Syt2. This number is in line with previous reported value for Syt1 at synapses of cultured hippocampal neurons (*Fernández-Alfonso et al., 2006*; *Wienisch and Klingauf, 2006*).

## Calmodulin-Munc13-1 signaling is crucial for regulating the coordinated uptake of vesicular proteins and membrane

The active zone protein Munc13-1 is an essential priming factor for exocytosis (*Varoqueaux et al., 2002*). The priming activity of Munc13s is regulated by three independent domains, one of which is a Ca$^{2+}$-CaM binding domain (*Junge et al., 2004*; *Lipstein et al., 2012*). Analyses of a knock-in (KI) mouse line that expresses a Ca$^{2+}$-CaM insensitive Munc13-1 variant (Munc13-1$^{W464R}$) instead of wild-type (WT) Munc13-1 (Munc13-1$^{WT}$) showed that the Ca$^{2+}$-CaM-Munc13-1 signaling pathway regulates the recovery rate of the releasable SV pool in the calyx of Held (*Lipstein et al., 2013*). Because some exocytotic proteins are also relevant for endocytosis, mainly by acting in the coupling between exo- and endocytosis, we hypothesized that the Ca$^{2+}$-CaM-Munc13-1 pathway might be an important post-exocytosis molecular organizer, in addition to its classical role in vesicle priming. Since CaM inhibition slowed the time course of Syt2 uptake, we considered the possibility of Munc13-1

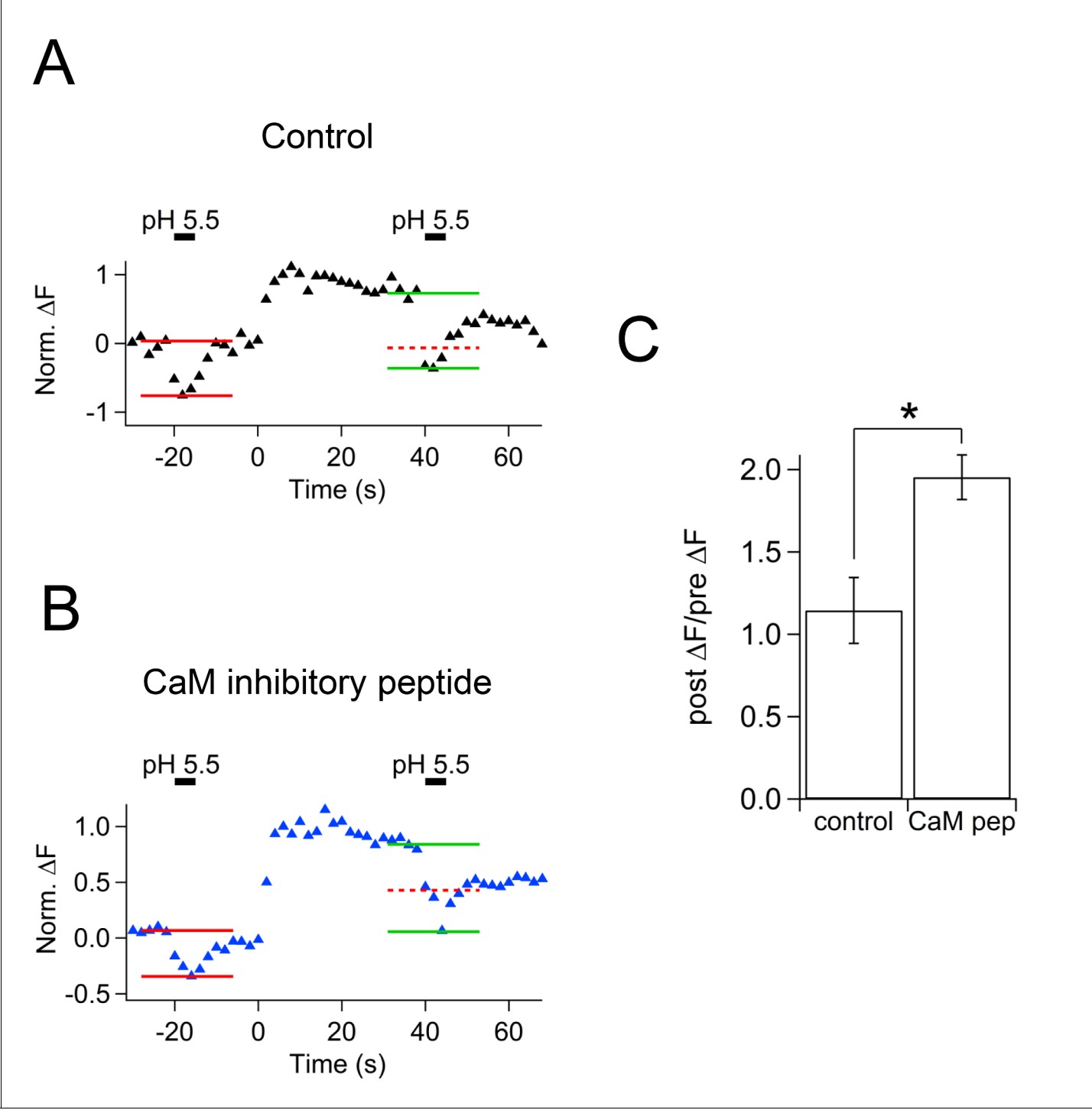

**Figure 6.** Effect of calmodulin inhibitory peptide on cypHer fluorescence changes induced by acidic solution application. (**A**) An example trace of the cypHer fluorescence change when an acidic extracellular solution (pH 5.5) was puff-applied during the recording without CaM inhibitory peptide. The pH 5.5 solution was puff applied for 5 s, once 20 s before and once 40 s after stimulation (50 ms pulse x 10). Red and green lines show the amount of fluorescence change induced by the pH 5.5 solution before and after stimulation, respectively. The dotted red line shows the amplitude of the 'before' signal. (**B**) The same as A, but in the presence of 20 μM CaM inhibitory peptide. (**C**) The de-quenching ratio before and after the stimulus (post ΔF/pre ΔF value) was larger in the presence of CaM inhibitory peptide (control, $1.15 \pm 0.20$, n = 5; CaM inhibitory peptide, $1.95 \pm 0.14$, n = 5, p=0.012).

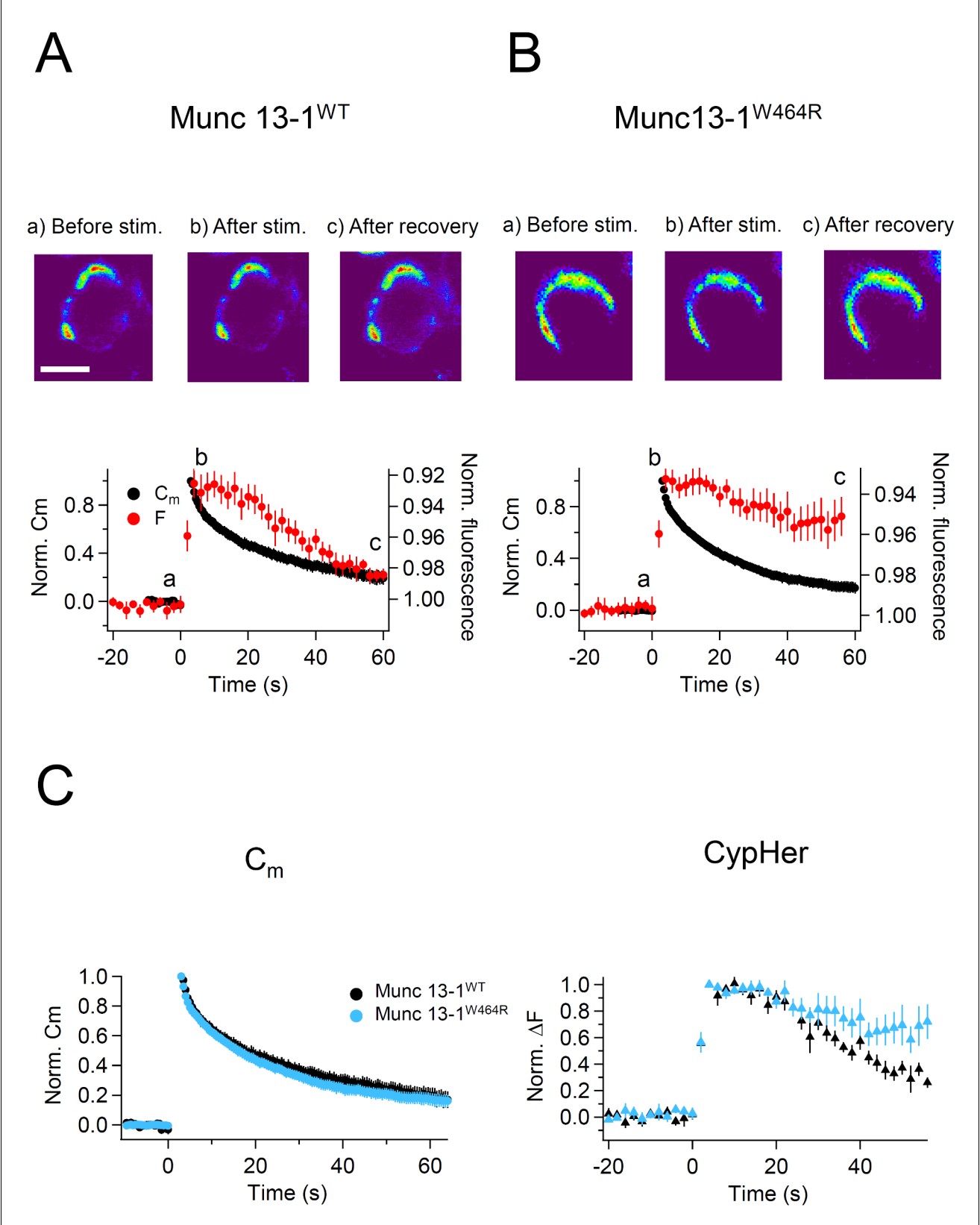

**Figure 7.** Capacitance and cypHer signal recovery in the calyx of Held of Munc 13-1$^{W464R}$ KI mice. (**A**) The same as *Figure 1D and E*, but recordings from calyx of Held terminals of wild-type (Munc13-1$^{WT}$) mice. The top panel shows the cypHer images at the time points shown in the bottom panel,
*Figure 7 continued on next page*

*Figure 7 continued*

and the bottom panel shows averaged $C_m$ (black circles) and cypHer fluorescence (red circles) changes (n = 8). Scale bar, 10 µm. (**B**) The same as A, but recordings from calyx of Held terminals of Munc 13-1$^{W464R}$ mice (n = 6). Black and red traces show average normalized $C_m$ and fluorescence changes. 3 out of 6 data were obtained from simultaneous measurements of capacitance and cypHer. (**C**) The left panel shows averaged $C_m$ traces from Munc13-1$^{WT}$ calyces (n = 8, black circles) and Munc13-1$^{W464R}$ calyces (n = 6, turquoise circles). The right panel shows averaged cypHer fluorescence traces from Munc13-1$^{WT}$ calyces (n = 8, black triangles) and Munc13-1$^{W464R}$ calyces (n = 6, turquoise blue triangles).

The following figure supplements are available for figure 7:

**Figure supplement 1.** Comparison of the recovery time course of capacitance and cypHer (mouse).

**Figure supplement 2.** Bleach correction for the cypHer fluorescence.

acting as a downstream target of CaM in this scenario, and tested if uptake of Syt2 was affected at calyx terminals of Munc13-1$^{W464R}$ KI mice as compared to WT littermates.

We used trains of ten 50 ms depolarizations to evoke exocytosis, and measured the recovery time courses of membrane capacitance and cypHer fluorescence. Recordings from WT mouse calyx terminals showed that the time course of cypHer signal recovery was similar to that of membrane capacitance retrieval (*Figure 7A*). Fits based on the assumption of re-acidification with an exponential time constant of 18.9 s after membrane retrieval and a 12 s delay of onset described the actual recovery of the cypHer signal very well (*Figure 7—figure supplement 1*). In contrast, recordings from Munc13-1$^{W464R}$ KI mouse calyx terminals showed a slower time course of cypHer signal recovery than of the membrane retrieval in 6 out of 7 recordings (*Figure 7B*). Comparison of averaged traces showed that the time course of membrane retrieval was not different between WT and Munc13-1$^{W464R}$ KI calyces (*Figure 7C* left, 56 s after the stimulus; Munc13-1$^{WT}$, 0.20 ± 0.05, n = 8; Munc13-1$^{W464R}$, 0.17 ± 0.04, n = 6, p=0.62), but the cypHer signal recovery was significantly smaller in Munc13-1$^{W464R}$ KI calyces as compared to WT calyces (*Figure 7C* right; Munc13-1$^{WT}$, 0.26 ± 0.04, n = 8; Munc13-1$^{W464R}$, 0.72 ± 0.13, n = 6, p=0.015). Together with the finding that the blockade of CaM inhibits the co-retrieval of Syt2 and membrane (*Figure 6*), this result indicates that the Ca$^{2+}$-CaM-Munc13-1 pathway is crucial for coordinated retrieval of Syt2 together with endocytosed membrane, supporting the notion that Munc13-1 is a downstream effector of CaM during endocytotic vesicular protein retrieval.

## Discussion

Coordinated cycling of vesicle membrane and vesicle proteins is a critical process by which the pool of releasable SVs is repopulated to maintain synaptic transmission over extended periods of time. Owing to the large size of the calyx of Held presynaptic terminal, we were able to simultaneously measure the time course of membrane and vesicular protein (Syt2) uptake in this synapse and made three key observations.

1. The vesicle membrane uptake, as assessed by capacitance measurements, and the Syt2 uptake, as assessed by the fluorescence of the anti-synaptotagmin2-cypHer5E probe, had a similar time course when slow endocytosis was elicited (*Figure 1*).
2. When fast endocytosis was elicited, Syt2 was still retrieved together with the membrane. However, acidification of the endocytosed organelles was very slow as determined by the divergence of the apparent recovery time courses of the membrane capacitance and the pH sensitive anti-synaptotagmin2-cypHer5E probe (*Figure 4*). This also suggests that pHluorin or related probes are not necessarily monitoring endocytosis faithfully.
3. The simultaneous uptake of membrane and Syt2 was disturbed by CaM inhibition (*Figure 5*), and the exocytosed Syt2 was left behind at the plasma membrane following membrane retrieval (*Figure 6*). In Munc13-1$^{W464R}$ calyces a similar phenotype was seen, indicating that Ca$^{2+}$-CaM-Munc13-1 signaling, which is triggered by Ca$^{2+}$ elevation during repetitive activity, might be crucial for regulating the coordinated retrieval of membrane and vesicular proteins (*Figure 7*).

On aggregate, these findings identify a signaling pathway that coordinates the retrieval of vesicle membrane and vesicle proteins, and that is regulated in a Ca$^{2+}$-CaM-Munc13-1-dependent manner.

Also, we suggest that fast mode of endocytosis may not be monitored properly using pHluorin based probes.

## Comparing the kinetics of vesicle membrane and vesicle protein retrieval

Several pH sensitive probes conjugated to vesicular proteins have been used to examine the kinetics of exo-endocytosis by taking advantage of the difference between intravesicular (~5.5) and extracellular (~7.4) pH. In many studies, it was assumed that the kinetics of the fluorescence changes represent endocytosis, including both, membrane and labeled protein retrieval. By directly comparing the kinetics of membrane and anti-Syt2-cypHer retrieval, we show here that this assumption is essentially correct at the calyx of Held presynaptic terminal when the stimulation intensity is mild. Under this condition, endocytosis is operates mainly by a slow, clathrin-dependent mode (*Yamashita et al., 2010*). The time course of cypHer signal recovery under this condition was slightly slower than that of the capacitance recovery, but the difference can be well described by taking the re-acidification time course (20 s ~ 40 s) into account. This is longer than recent reports showing that the re-acidification time course of glutamatergic vesicles takes 15 s in cultured hippocampal neurons (*Egashira et al., 2015*). In some studies, even faster re-acidification time courses were suggested (*Gandhi and Stevens, 2003*; *Atluri and Ryan, 2006*). One of the possibilities that may cause the slow re-acidification time course was relatively large size of the synaptic vesicles in the calyx terminal (50 nm, *Sätzler et al., 2002*). These vesicles may contain up to 5,000–7,000 glutamate molecules. Recently, *Cho and von Gersdorff. (2014)* reported that vesicular glutamate may function as a buffer for vesicle protons. So a larger vesicle may contain many hundreds to thousands of protons and its acidification may take a relatively long time. However, the time constant for single vesicle glutamate filling at the calyx of Held is estimated to be 15 s (*Hori and Takahashi, 2012*). In our train stimulation protocol, 37% of membrane retrieval was mediated by fast endocytosis, thus larger endosome-like structure might also contribute to the slower re-acidification time course. It may be possible that Syt2 remains stranded in the plasma membrane for ~15 s and then a partial contribution of larger endosome-like structure is needed (bulk endocytosis) to explain the 38.7 s decay time constant (*Figure 1—figure supplement 2*).

Previous reports employing pHluorin-based probes showed that endocytosis gets slower with stronger stimulation (*Armbruster et al., 2013*; *Fernández-Alfonso and Ryan, 2004*). Usually, clathrin-dependent, slow endocytosis is observed after mild stimulation, but in these pHluorin-based studies (*Armbruster et al., 2013*; *Fernández-Alfonso and Ryan, 2004*), endocytosis-based signal recovery was still slow (tens of seconds) even with strong stimulation, which elicits clathrin-independent endocytosis. In contrast, capacitance measurements and electron microscopic studies showed that clathrin-independent endocytosis, i.e. either bulk endocytosis or ultrafast endocytosis, occurs in the range of tens of ms to seconds (*Jockusch et al., 2005*; *Wu et al., 2005*; *Watanabe et al., 2013*). Fast endocytosis in the present study (*Figure 4*) reflects clathrin-independent endocytosis (*de Lange et al., 2003*), and by providing a direct comparison of membrane uptake, vesicle protein uptake, and re-acidification of endocytosed organelles, our results reconcile the previously reported, apparent discrepancy in the kinetics of endocytosis as assessed by pHluorin probes vs. capacitance measurements and electron microscopic analyses. Our data demonstrate that the retrieval of vesicle membrane and vesicle proteins occurs simultaneously, but that re-acidification of the endocytosed organelle is slow after clathrin-independent, fast endocytosis. This is likely due to the large surface-to-volume ratio of the endosome-like structures that form during fast endocytosis and/or to slow budding of vesicles from such endosomes (*Watanabe et al., 2014*; *Kononenko et al., 2014*; *Kononenko and Haucke, 2015*)

## The molecular basis of coordinated retrieval of vesicle membrane and vesicle proteins

In the present study, we examined the exo-endocytotic cycling of Syt2 using cypHer as a reporter. The lack of SV2 (*Yao et al., 2010*) or stonin2 (*Kononenko et al., 2013*) compromises the fidelity of synaptotagmin sorting, but the kinetics of clathrin-dependent slow endocytosis is unchanged even when synaptotagmin sorting is perturbed (*Kononenko et al., 2013*). This suggests that membrane retrieval and sorting of SV proteins can be segregated. Our results are consistent with this notion,

and the two processes can be separated in the presence of a CaM inhibitor or in Munc13-1[W464R] calyces (*Figures 5*, *7*). However, the possibility remains that several independent cycling pathways exist for a given vesicle protein. For instance, VGLUT deficiency slows the recycling of synaptophysin but not of Syt1 (*Pan et al., 2015*), while stonin 2 deficiency slows the recycling of Syt1 but not of synaptophysin or synaptobrevin 2 (*Kononenko et al., 2013*).

CaM inhibitors and the perturbation of $Ca^{2+}$-CaM-Munc13-1 signaling perturb Syt2 uptake without altering membrane retrieval kinetics (*Figures 5*, *7*). However, calcineurin, a downstream target of CaM, was shown to be necessary for membrane retrieval (*Sun et al., 2010*; *Wu et al., 2014*). For three reasons, our results do not necessarily contradict these calcineurin data. First, it is possible that the complex between CaM and calcineurin is so tight that higher concentrations of CaM inhibitors are required to block the function of calcineurin in membrane retrieval. Second, the effect of calmodulin blockers on membrane retrieval can be seen when low concentrations of $Ca^{2+}$ buffers are present in the presynaptic patch pipette, which lead to a strong elevation of the global presynaptic calcium concentration (*Wu et al., 2014*), but not when high concentrations of $Ca^{2+}$ buffers are used as in the present study (see Materials and methods). Third, a CaM-independent component of membrane retrieval might exist. All this notwithstanding, our results suggest that CaM has several downstream targets that regulate the retrieval of membrane and proteins.

Clathrin-independent endocytosis, which forms large endosome-like structures by retrieving large pieces of membrane, has been observed at many types of synapses after strong stimulation (*Thomas et al., 1994*; *Holt et al., 2003*; *Wu et al., 2009*). Recent findings indicate that the plasma membrane adaptor AP-2 is required for vesicle regeneration from bulk endosome-like organelles in central nerve terminals (*Kononenko et al., 2014*; *Watanabe et al., 2014*), but it is not known how long it takes for vesicles to bud from the endosome-like organelles after bulk endocytosis. We found in this context that re-acidification barely occurs 1 min after a 2 s depolarization (*Figure 4*), which suggests that vesicle budding from endosome-like organelles is slow (*Cheung et al., 2010*). Our results also suggest that Syt2 is retrieved together with the membrane not only in the slow, but also in the fast mode of endocytosis, but the molecular mechanism of Syt2 sorting during bulk endocytosis remains to be elucidated.

## The role of CaM-Munc-13 signaling in the coordinated retrieval of membrane and Syt2

We showed that CaM inhibition slowed down Syt2 retrieval without affecting the kinetics of membrane retrieval (*Figures 5*, *6*), and our data indicate that Munc13-1 is a possible downstream target of $Ca^{2+}$-CaM signaling in this process (*Figure 7*). This is somewhat surprising since Munc13-1 is a well-known vesicle priming factor at the active zone. In this context, the $Ca^{2+}$-CaM-Munc13-1 signaling pathway regulates the recovery rate of the pool of releasable SVs in hippocampal neurons (*Junge et al., 2004*; *Lipstein et al., 2012*) and in the calyx of Held terminal (*Lipstein et al., 2013*), likely by enhancing the replenishment rate of synaptic vesicles. However, the effect of dynamin inhibition, which perturbs endocytosis, on the SV replenishment rate (*Wu et al., 2009*; *Hosoi et al., 2009*) is very similar to the effect measured in the Munc13-1[W464R] KI calyces (*Figure 7*), and to that of acute pharmacological blockade of CaM (*Sakaba and Neher, 2001*). Because the effects of perturbed endocytosis on vesicle pool recovery can be explained by delayed clearance of active zone release sites from the remains of molecular complexes formed by the preceding fusion (*Hosoi et al., 2009*; *Kawasaki et al., 2000*), or by impaired structural recovery after the preceding exocytosis (*Wu et al., 2009*) in addition to priming of synaptic vesicles (*Midorikawa and Sakaba, 2015*), and based on the results of our present study, we propose that the $Ca^{2+}$-CaM-Munc-13 complex does not only act in the regulation of SV replenishment but is also involved in release site clearance. The corresponding roles of the $Ca^{2+}$-CaM-Munc-13 complex might be dependent on or independent of each other. The total amount of exocytosis evoked by a train of ten 50 ms stimuli was smaller in Munc13-1[W464R] KI calyces as compared to WT controls, but the amount of exocytosis elicited by the first pulse in the train was not different (data not shown), which is consistent with a previous study showing slower SV replenishment at release sites in Munc13-1[W464R] KI calyces (*Lipstein et al., 2013*).

Our finding that the uptake of Syt2 is perturbed by CaM inhibition and in Munc13-1[W464R] calyces supports the idea that $Ca^{2+}$-CaM-Munc13-1 signaling plays a role in endocytosis, and particularly in the retrieval of Syt2. Disruption of exocytotic proteins such as SNAREs affects endocytosis, which

suggests a close coupling between exo- and endocytotic processes (*Hosoi et al., 2009*; *Zhang et al., 2013*). Syt1 is needed for $Ca^{2+}$-dependence of clathrin-mediated endocytosis in chromaffin cells (*Yao et al., 2012*; *McAdam et al., 2015*), and inhibition of Syt2 interaction to AP2 blocks clathrin-mediated endocytosis in calyx terminals (*Hosoi et al., 2009*). Also, genetic deletion of SV proteins perturbs endocytosis (*Nicholson-Tomishima and Ryan, 2004*; *Deák et al., 2004*). One possible molecular scenario is that the core complex of the release machinery, composed of SNAREs, Munc13s, Munc18s, and synaptotagmin (*Betz et al., 1997*; *Ma et al., 2013*), has to be disassembled before endocytosis, and that without a functional $Ca^{2+}$-CaM-Munc13-1 pathway, exocytosed Syt2 might be harder to dissociate from the release complex, so that Syt2 translocation to the endocytotic sites is retarded, causing a molecular jam at the active zone. It will be interesting to examine whether the uptake of other proteins involved in the vesicle fusion complex are also regulated by $Ca^{2+}$-CaM-Munc13-1 signaling.

In any case, our results show that the coordinated endocytotic retrieval of membrane and proteins is subject to modulation by second messenger pathways, and thus is a potential target of modulation during presynaptic plasticity. This might be particularly relevant in the calyx of Held, which has to cope with high-frequency transmission (*Taschenberger et al., 2002*) and where $Ca^{2+}$ elevation boosts proper SV sorting and replenishment, but other synapses may employ the same regulatory mechanism. So far, the calyx of Held terminal is the most convenient neuronal presynaptic terminal for capacitance measurements, but recently several other neuronal presynaptic terminals, where genetic manipulations are better feasible, were shown to be amenable to capacitance recordings (e.g. hippocampal mossy fiber terminals, *Hallermann et al., 2003*; cerebellar mossy fiber boutons, *Delvendahl et al., 2015*; cultured cerebellar Purkinje cell, *Kawaguchi and Sakaba, 2015*). Because endocytotic mechanisms may differ among synapse types (*Kononenko and Haucke, 2015*), it will be interesting to examine the differential kinetics of vesicular protein retrieval in these preparations as well.

## Materials and methods

### Ethical approval

Animal care and animal procedures were conducted in accordance with the guidelines of the Physiological Society of Japan, and were approved by the Doshisha University Committee for Regulation on the Conduct of Animal Experiments and Related Activities. All efforts were taken to minimize animal numbers. The generation, maintenance, and use of the Munc13-1[W464R] mice were approved by the responsible local government organization (Niedersächsisches Landesamt für Verbraucherschutz und Lebensmittelsicherheit, permissions 33.9.42502-04-13/1359 and 33.19-42502-04-15/1817).

### Electrophysiology

Transverse brainstem slices (200 μm thickness) were prepared from Wistar rats, C57BL6 mice, and Munc 13-1[W464R] KI mice (P9-11 each) using a Leica VT1200S slicer (Leica Microsystems, Wetzlar, Germany). During slicing, the brainstem was kept in ice-cold solution containing (in mM) 130 Sucrose, 60 NaCl, 2.5 KCl, 25 glucose, 25 $NaHCO_3$, 1.25 $NaH_2PO_4$, 0.5 ascorbic acid, 3 myoinositol, 2 Na-pyruvate, 0.1 $CaCl_2$, and 3 $MgCl_2$. In order to allow the slices to recover from the cutting procedure, slices were incubated at 37°C for 1 hr in a standard extracellular solution containing (in mM) 125 NaCl, 2.5 KCl, 25 glucose, 25 $NaHCO_3$, 1.25 $NaH_2PO_4$, 0.4 ascorbic acid, 3 myoinositol, 2 Na-pyruvate, 2 $CaCl_2$, and 1 $MgCl_2$ (pH 7.4, gassed with 95% $O_2$ and 5% $CO_2$). Slices were visualized on an upright microscope (Axioskop; Zeiss, Oberkochen, Germany). The calyx of Held presynaptic terminals were whole-cell voltage clamped at -80 mV using an EPC9/2 amplifier (HEKA, Lambrecht, Germany) controlled by PatchMaster software (HEKA). The presynaptic patch pipettes (7–10 MΩ) were filled with intracellular solution containing (in mM) 140 Cs-gluconate, 20 TEA-Cl, 10 HEPES, 5 $Na_2$-phosphocreatine, 4 MgATP, 0.3 NaGTP, and 0.5 EGTA (pH 7.3). When using calmodulin/dynamin inhibitory peptide, reagents were added to the intracellular solution. For recordings of wild type mice, some data (5 out of 8) were taken with CsCl based intracellular solution, in which Cs-gluconate was replaced by CsCl. Since we found no difference by using CsCl solution, we pooled the corresponding data.

The presynaptic series resistance (10–25 MΩ) was compensated by 10–50% as appropriate. During recordings, 1 μM TTX and 10 mM TEA-Cl were included in the extracellular solution to block

Na$^+$ and K$^+$ channels respectively, and presynaptic Ca$^{2+}$ currents were isolated. Only cells with stable membrane resistance (Rm), leak current below 50 pA at holding potential (-80 mV) and stable series resistance below 25 MΩ were considered in the study.

Membrane capacitance measurements from the calyx of Held presynaptic terminals were performed using an EPC9/2 amplifier in the sine+DC configuration (*Lindau and Neher, 1988*). A sine wave (30 mV in amplitude, 1000 Hz in frequency) was superimposed on a holding potential of -80 mV. Experiments were performed at room temperature. TTX and bafilomycin were obtained from Wako (Osaka, Japan). ATP, GTP, and TEA-Cl were obtained from Sigma-Aldrich (St. Louis, Missouri). The calmodulin and dynamin inhibitory peptides were from Calbiochem (Darmstadt, Germany). Other reagents were from Nacalai tesque (Kyoto, Japan).

### Immunostaining

CypHer5E dye conjugated to antibodies directed against the luminal domain of Syt2, anti-Syt2-cypHer (Synaptic Systems, Göttingen, Germany), was used to monitor exo-endocytosis from the calyx presynaptic terminal. To label the presynaptic terminals, slices were incubated with anti-Syt2-cypHer at 37°C for 30 min in a high K$^+$ extracellular solution containing (in mM) 95 NaCl, 32.5 KCl, 25 glucose, 25 NaHCO$_3$, 1.25 NaH$_2$PO$_4$, 0.4 ascorbic acid, 3 myoinositol, 2 Na-pyruvate, 2 CaCl$_2$, and 1 MgCl$_2$ (pH 7.4). Subsequently, slices were held at 37°C for up to 3 hr in the standard extracellular solution until mounting onto the microscope.

### Fluorescence imaging

Experiments were performed at room temperature on an upright microscope (Axioskop, Zeiss) equipped with a 60x, 0.9 NA water-immersion objective (Olympus, Tokyo, Japan). Images (1344 x 1024 pixels) were acquired with a CCD camera (ORCA-R2 Digital CCD camera C10600; Hamamatsu Photonics, Shizuoka, Japan) controlled by HoKaWo software (Hamamatsu Photonics). CypHer was excited at 645 nm with a monochromator (Polychrome V; Till Photonics, Hillsboro, Oregon) triggered by PatchMaster software (HEKA) and imaged using a 692/40 nm single-band bandpass filter (Semrock, Rochester, New York). Time-lapse images were acquired at 0.5 Hz with 300–500 ms exposure time. Time-lapse images were corrected for photobleaching by subtracting the bleaching time course of the neighboring calyx terminal (*Figure 7—figure supplement2*). The fluorescence intensity of the region of interest was background-subtracted, and normalized either to initial intensity or to ΔF induced by stimulation.

To monitor fluorescence changes during exo-endocytosis, the calyx of Held terminals were stimulated with a train of depolarizing pulses (+70 mV for 2 ms followed by repolarization to 0 mV for 50 ms, repeated ten times with an inter-stimulus interval of 200 ms) and a step depolarizing pulse (+70 mV for 2 ms followed by repolarization to 0 mV for 2 s). For experiments involving the application of acidic solutions, MES-buffered (25 mM) extracellular solution at pH 5.5 was puff-applied to target terminals using a Pneumatic PicoPump (World Precision Instruments, Sarasota, Florida) before and after stimulation. The MES-buffered solution was made by replacing 25 mM NaHCO$_3$ of the standard external solution to 25 mM MES. The puff pipette (4–6 MΩ) was positioned ∼100 μm away from the target terminal.

### The Munc 13-1$^{W464R}$ KI mice

Munc 13-1$^{W464R}$ KI mutant mice were generated as described previously (*Lipstein et al., 2013*). In these mice, a point mutation in exon 11 of the *Unc13a* gene replaces the tryptophane in position 464 of Munc13-1 by an arginine and produces a Munc13-1 mutant that does not bind CaM (*Junge et al., 2004*; *Lipstein et al., 2013*). Wild type littermates (Munc13-1$^{WT}$) were used as controls, and the genotypes of the mice were determined by PCR before and after the experiments.

### Image and data analysis

Images and data were analyzed using IGOR Pro 6 (WaveMetrics, Lake Oswego, Oregon) and Excel 2013 software (Microsoft, Redmond, Washington). All values are given as mean ± SEM. Statistical significance was determined by Student's t test. p values smaller than 0.05 were considered to indicate statistically significant differences.

When cypHer fluorescence was fitted by assuming a certain delay and a re-acidification time constant of the endocytosed organelle after membrane retrieval (*Figure 1—figure supplement 2*, *Figure 7—figure supplement 1*), the pH-dependence of the cypHer fluorescence was described by a Henderson-Hasselbalch equation with a Hill coefficient of 1, and a pKa of 7.05 (*Hua et al., 2011*).

## Acknowledgements

We thank I. Herfort for the technical assistance and Dr. S Kawaguchi for the helpful comments. This work was supported by JSPS/MEXT KAKENHI Grant Numbers 15H04261, 15K14321, 26110720 to TS, 15K18346 to MM, Core-to-Core Program A Advanced Research Networks, the Toray Science Foundation (TS), the Uehara Foundation (TS), and by an ERC Advanced Grant of the European Union (NB).

## Additional information

### Funding

| Funder | Grant reference number | Author |
| --- | --- | --- |
| JSPS/MEXT KAKENHI | 15H04261 | Takeshi Sakaba |
| Core-to-Core Program A | | Takeshi Sakaba |
| Torey Science Foundation | | Takeshi Sakaba |
| Uehara Memorial Foundation | | Takeshi Sakaba |
| ERC Advanced Grant of the European Union | | Nils Brose |
| JSPS/MEXT KAKENHI | 15K14321 | Takeshi Sakaba |
| JSPS/MEXT KAKENHI | 26110720 | Takeshi Sakaba |
| JSPS/MEXT KAKENHI | 15K18346 | Mitsuharu Midorikawa |

The funders had no role in study design, data collection and interpretation, or the decision to submit the work for publication.

### Author contributions

YO, Acquisition of data, Analysis and interpretation of data, Drafting or revising the article, Contributed unpublished essential data or reagents; NL, NB, Drafting or revising the article, Contributed unpublished essential data or reagents; YH, K-HL, Acquisition of data, Drafting or revising the article, Contributed unpublished essential data or reagents; TS, Conception and design, Drafting or revising the article, Contributed unpublished essential data or reagents; MM, Conception and design, Acquisition of data, Analysis and interpretation of data, Drafting or revising the article, Contributed unpublished essential data or reagents

### Author ORCIDs

Mitsuharu Midorikawa, http://orcid.org/0000-0003-4818-6986

### Ethics

Animal experimentation: Animal care and animal procedures were conducted in accordance with the guidelines of the Physiological Society of Japan, and were approved by the Doshisha University Committee for Regulation on the Conduct of Animal Experiments and Related Activities. All efforts were taken to minimize animal numbers. The generation, maintenance, and use of the Munc13-1W464R mice were approved by the responsible local government organization (Niedersächsisches Landesamt für Verbraucherschutz und Lebensmittelsicherheit, permissions 33.9.42502-04-13/1359 and 33.19-42502-04-15/1817).

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
