## [Decision Letter]

Thank you for submitting your article "Distinct modes of endocytotic presynaptic membrane and protein uptake at the calyx of Held terminal" for consideration by *eLife*. Your article has been reviewed by two peer reviewers, and the evaluation has been overseen by a Reviewing Editor and Gary Westbrook as the Senior Editor. The reviewers have discussed the reviews with one another and the Reviewing Editor has drafted this decision to help you prepare a revised submission.

Summary:

All reviewers thought that this manuscript is important and exciting, comparing for the first time the kinetics of compensatory endocytosis of membrane and proteins in calyx of Held nerve terminals and two independent techniques: time-resolved membrane capacitance and pHluorin-mediated fluorescent imaging of anti-Syt2-cypHer. In addition, the manuscript provides important molecular mechanisms that decouple protein and membrane recycling. However, several questions related to the data analysis were raised, and that need to be addressed. Also, the discussion related to acidification estimates of synaptic vesicles requires attention.

Essential revisions:

Analysis:

The analysis of the time course of the cypHer5E fluorescence changes is not clearly described and the statements on the time course appear unjustified. First, the concept of a time constant for the fluorescence recovery time course seems not justified because the analyzed average time course (Figure 1—figure supplement 2, red line) does not show evidence of a decreasing slope, instead the slope is apparently rather constant from 15 s to 60 s. Figure 1 (top and bottom panels) suggest that the decrease is still continuing at 60 s. Has the fluorescence generally been recorded for longer times (as in Figure 4—figure supplement 1) to show that there is actually an asymptotic value? Figure 1—figure supplement 1, right, suggests that the recovery is sometimes only partial. Are longer recordings available? Do the authors assume that the decline in fluorescence (Figure 1, top) returns to the pre-stimulation baseline? If so, how can this assumption be justified?

Figure 1—figure supplement 2 shows exponential decays – why are they noisy? It appears that the capacitance data has somehow been used to generate these curves but there is not a corresponding description. Is it a convolution of the C_m_ time course with a re-acidification time constant? If so, it would be helpful to actually fit this time constant (least squares). If the bold orange line is the best fit it would suggest that the hypothesis might be wrong because the best match (bold orange) does not seem to be a good fit, a straight line would do much better. The analysis needs clarification and/or further analysis.

2) Can it be concluded that the recovery (Figure 3) is slower? Apparently the slow phase of the C_m_ trace and the fluorescence recovery are absent with the dynamin inhibitor and the fluorescence recovery is also absent in the bafilomycin experiment.

3) The issue if the recovery is slower or its extent smaller also applies to the text referring to other experiments.

4) The authors seem to assume that fast endocytosis in the bulk endocytosis mode after the 2 s pulses is the same mechanism as that of the fast phase after short pulses. What is the evidence for that?

5) Recent experiments have shown the in chromaffin cells Syt1 is needed for Ca-dependence of clathrin-mediated endocytosis and that it involves dynamin. How do the authors relate their findings to these results (Yao et al. 2012 J Neurosci, McAdam et al. 2015 J Cell Sci)? Should one expect a similar role for Syt2 in the calyx?

Discussion:

Atluri and Ryan are quoted several times saying that "…acidification of glutametergic SVs occurs with a time constant of tens of seconds." But this is not very accurate. Atluri and Ryan (JNeurosci., 2006) estimated a time constant of 4 -5 seconds (at room temperature) and Gandhi and Stevens (Nature, 2003) estimated a time constant of 0.4 seconds. So the time constant of 30 seconds after a 10 s delay that the authors estimate is very surprising and seems very slow. On the other hand, synaptic vesicles in the calyx of Held are quite large (50 nm diameter) and may contain up to 5000-7000 glutamate molecules. Recently, Cho and v. Gersdorff (JNeurosci., 2014) reported that vesicular glutamate may function as a buffer for vesicle protons. So a larger vesicle may contain many hundreds to thousands of protons and its acidification may take a relatively long time. However, the time constant for single vesicle glutamate filling at the calyx of Held is estimated to be 15 seconds (Hori and Takahashi, Neuron, 2012). Please comment on this previous result. In conclusion, it seems that Syt2 remains stranded in the plasma membrane for 10 s and then a larger endosome-like structure is needed (bulk endocytosis) to explain the very slow 30 s decay time constant. All these references should be briefly discussed and cited in the Discussion and the authors should mention in the Abstract or Introduction this surprising finding of the 10 s delay followed by a 30 s decay time constant for reacidification.

---

## [Author Response]

Essential revisions:

Analysis:

The analysis of the time course of the cypHer5E fluorescence changes is not clearly described and the statements on the time course appear unjustified. First, the concept of a time constant for the fluorescence recovery time course seems not justified because the analyzed average time course (Figure 1—figure supplement 2, red line) does not show evidence of a decreasing slope, instead the slope is apparently rather constant from 15 s to 60 s.

Thank you for the detailed consideration of the analysis. We added 3 more cells for the cypHer fluorescence data, and now fitting become better. The new trace was best fitted using least squares method with 38.7s re-acidification time constant with a 14s delay (Figure 1—figure supplement 2). In addition, to show variability of the data, we added error bars to the cypHer trace in Figure 1—figure supplement 2.

Figure 1 (top and bottom panels) suggest that the decrease is still continuing at 60 s. Has the fluorescence generally been recorded for longer times (as in Figure 4—figure supplement 1) to show that there is actually an asymptotic value?

We also added longer recording to show that the cypHer fluorescence showed decreasing slope after longer recording period (Figure 1—figure supplement 3). As the reviewers mentioned, the decrease is still continuing to some extent at 60 s but the slope becomes shallower afterwards, and can be well fitted by an exponential. The slopes until 60 s were similar between shorter recordings and longer recordings (Figure 1—figure supplement 3). The exponential fitting to the longer cypHer recording gives asymptotic value of 0.13, which is similar to the asymptotic value of 0.12 calculated from fitting the averaged C_m_ recording (shown in Figure 1) with a double exponential, as described in the first paragraph of the subsection “Simultaneous recordings of membrane capacitance and anti-Syt2-cypHer uptake”.

We added this point as “The asymptotic value (extrapolated from the exponential fit) of the fluorescence recovery (0.13 from Figure 1—figure supplement 3) was similar to that of the capacitance trace (0.12 from Figure 1), which indicated that the rate of endocytosis becomes as low as that of re-acidification. The exponential decay time constant was more clearly seen by taking longer recording (Figure 1—figure supplement 3), suggesting that exocytosed Syt2 is almost completely retrieved.”.

Figure 1—figure supplement 1, right suggests that the recovery is sometimes only partial. Are longer recordings available?

The partial recovery observed in Figure 1—figure supplement 1 is probably due to photobleaching. In Figure 1—figure supplement 1, because we used field stimulation to elicit exocytosis, which stimulates all the calyx, we could not use neighboring non-stimulated calyx terminal as a reference as in Figure 7—figure supplement 2, hence photobleaching was not corrected. In Figure 7—figure supplement 2 figure legend, we added “The apparent partial recovery is due to photobleaching (Hua et al., 2011).” Subtracting the baseline decline before stimulation also suggests that incomplete recovery is likely to be due to photobleaching (please see Figure 8 ).

Author response image 1.**DOI:**
http://dx.doi.org/10.7554/eLife.14643.016

Do the authors assume that the decline in fluorescence (Figure 1, top) returns to the pre-stimulation baseline? If so, how can this assumption be justified?

As seen in the longer recording (Figure 1—figure supplement 3), we think that the decline in fluorescence returns to pre-stimulation baseline, though there may be a possibility that a small fraction (10%) may remain at the plasma membrane. However, we should note that our analysis does not assume that fluorescence returns to baseline, and for the fitting, the baseline value is a free parameter.

Figure 1—figure supplement 2 shows exponential decays – why are they noisy? It appears that the capacitance data has somehow been used to generate these curves but there is not a corresponding description. Is it a convolution of the C_m_ time course with a re-acidification time constant? If so, it would be helpful to actually fit this time constant (least squares). If the bold orange line is the best fit it would suggest that the hypothesis might be wrong because the best match (bold orange) does not seem to be a good fit, a straight line would do much better. The analysis needs clarification and/or further analysis.

As the reviewers pointed out, the fitting was a convolution of the C_m_ time course with a delay and a re-acidification time constant of the endocytosed organelle. We described the fitting procedure more clearly both in figure legends and Methods section, and performed the fitting by least squares method. We changed the Figure 1—Figure 2 and Figure 7—Figure 1 figure legends as “By assuming a certain delay and a re-acidification time constant of the endocytosed organelle after membrane retrieval, the recovery time course of cypHer signal (red circles) was fitted. […] During the delay, the value was held to be 1. The cypHer signal was best fitted with a 14 (13) s delay with a 38.7 (18.9) s re-acidification time constant after membrane retrieval (dotted line).”

In the Methods section, we also added “When cypHer fluorescence was fitted by assuming a certain delay and a re-acidification time constant of the endocytosed organelle after membrane retrieval (Figure 1—figure supplement 2, Figure 7—figure supplement 1), the pH-dependence of the cypHer fluorescence was described by a Henderson-Hasselbalch equation with a Hill coefficient of 1, and a pKa of 7.05 (Hua et al., 2011).” By taking longer recordings, now we show that control fluorescence shows decreasing slope (Figure 1—figure supplement 3), and we think the fitting become more meaningful.

2) Can it be concluded that the recovery (Figure 3) is slower? Apparently the slow phase of the C_m_ trace and the fluorescence recovery are absent with the dynamin inhibitor and the fluorescence recovery is also absent in the bafilomycin experiment.

Thank you very much for suggesting better description of the result. From the dynamin inhibitor experiment, we concluded that the slow phase of membrane retrieval and fluorescence recovery was blocked. Fast phase of membrane retrieval still remained, but fluorescence recovery was negligible probably because the fast endocytosis was mediated by bulk endocytosis and their re-acidification was very slow. Instead of using the word “slower”, now we describe the effect as “blocked”, and change the manuscript as “Both the capacitance and the cypHer signal recovery were blocked, and their recovery became significantly smaller in the presence of dynamin inhibitory peptide than under control conditions”.

In the bafilomycin experiment section, we also described the effect as “blocked” instead of “slower”, and changed the sentence as follow “but the recovery of cypHer signal was blocked with bafilomycin”.

3) The issue if the recovery is slower or its extent smaller also applies to the text referring to other experiments.

As for CaM inhibitory peptide experiment, we changed the description as “but the cypHer signal recovery was reduced and/or slowed down by application of CaM inhibitory peptide”. In Munc13-1^W464R^ KI experiment, we also changed the description as “but the cypHer signal recovery was significantly smaller in Munc13-1^W464R^ KI calyces as compared to WT calyces”.

4) The authors seem to assume that fast endocytosis in the bulk endocytosis mode after the 2 s pulses is the same mechanism as that of the fast phase after short pulses. What is the evidence for that?

The mode of endocytosis is stimulus-dependent, and it has been shown that the fast mode of endocytosis is seen in response to strong stimulation in P8-11 calyx of Held terminals (Wu et al., 2005; Yamashita et al., 2005, 2010). With similar train stimulation protocol that we used in this work (20ms x 10 depolarization at 10 Hz), Wu et al. (2005) reported that about 35% of the endocytosis was mediated by fast mode of endocytosis, which is consistent with our result (37% fast endocytosis), and Renden and von Gersdorff (2007), Wu et al. (2009) and Yamashita et al. (2010) showed that the amount of fast endocytosis increased gradually with stronger stimulation. In Midorikawa et al. (2014), we also showed that the amount of fast mode of endocytosis was gradually increased by elongating pulse durations in P8-11 calyx of Held terminals, and fast endocytosis after shorter (200 ms) pulse and longer (500 ms) pulse were both P/Q type Ca channel dependent. Also, fast mode of endocytosis at the calyx seems to be insensitive to dynamin inhibitions (Figure 3, Yamashita et al., 2010; but see Xue et al., 2008), suggesting the same mechanism. We emphasized the progressive stimulus dependence of the fast endocytosis by adding the sentence “and that the contribution of fast endocytosis increased progressively by applying stronger stimulation (Renden and von Gersdorff, 2007; Wu et al., 2009; Yamashita et al., 2010; Midorikawa et al., 2014).” in Introduction section, and “It has been shown that the amount of fast mode of endocytosis was gradually increased by elongating pulse durations in P8-11 calyx of Held terminals (Renden and von Gersdorff, 2007;Wu et al., 2009; Yamashita et al., 2010; Midorikawa et al., 2014).” in Results section. We also added “This fast mode of endocytosis is shown to be insensitive to dynamin inhibition (Yamashita et al., 2010, but see Xue et al., 2008;), as in Figure 3 fast component, suggesting the same mechanism.” to clearly show that fast endocytosis in the bulk endocytosis mode after the 2 s pulses and that of the fast phase after short pulses share the same mechanism.

5) Recent experiments have shown the in chromaffin cells Syt1 is needed for Ca-dependence of clathrin-mediated endocytosis and that it involves dynamin. How do the authors relate their findings to these results (Yao et al. 2012 J Neurosci, McAdam et al. 2015 J Cell Sci)? Should one expect a similar role for Syt2 in the calyx?

Thank you for pointing out the important references. We cited both works in the manuscript. Because it has been demonstrated that Ca^2+^ is critical for clathrin-mediated endocytosis (CME), the slow form of endocytosis, in the calyx of Held, and that inhibition of Syt2 interaction to AP2 by the inhibiting peptide blocks CME in the calyx of Held (Hosoi et al., 2009), it is possible that Syt2 is needed for CME. We mention these studies in a Discussion section as “Syt1 is needed for Ca-dependence of clathrin-mediated endocytosis in chromaffin cells (Yao et al., 2012; McAdam et al., 2015), and inhibition of Syt2 interaction to AP2 blocks clathrin-mediated endocytosis in calyx terminals (Hosoi et al., 2009).” However, we cannot tell if Syt2 is needed for Ca-dependence of clathrin-mediated endocytosis from our data.

Discussion:

Atluri and Ryan are quoted several times saying that "…acidification of glutametergic SVs occurs with a time constant of tens of seconds." But this is not very accurate. Atluri and Ryan (JNeurosci., 2006) estimated a time constant of 4-5 seconds (at room temperature) and Gandhi and Stevens (Nature, 2003) estimated a time constant of 0.4 seconds. So the time constant of 30 seconds after a 10 s delay that the authors estimate is very surprising and seems very slow. On the other hand, synaptic vesicles in the calyx of Held are quite large (50 nm diameter) and may contain up to 5000-7000 glutamate molecules. Recently, Cho and v. Gersdorff (JNeurosci., 2014) reported that vesicular glutamate may function as a buffer for vesicle protons. So a larger vesicle may contain many hundreds to thousands of protons and its acidification may take a relatively long time. However, the time constant for single vesicle glutamate filling at the calyx of Held is estimated to be 15 seconds (Hori and Takahashi, Neuron, 2012). Please comment on this previous result. In conclusion, it seems that Syt2 remains stranded in the plasma membrane for 10 s and then a larger endosome-like structure is needed (bulk endocytosis) to explain the very slow 30 s decay time constant. All these references should be briefly discussed and cited in the Discussion and the authors should mention in the Abstract or Introduction this surprising finding of the 10 s delay followed by a 30 s decay time constant for reacidification.

Thank you very much for pointing out the important references and the suggestions to add as discussion points. We quote Atluri & Ryan (2006) and Gandhi & Stevens (2003) as examples of short reacidification times, and cite Hori & Takahashi (2012) and Egashira et al. (2015) for longer reacidification times. We mentioned that the reacidification time constant of 40 s observed in rat calyx terminal was slow, and mentioned the large size and the resultant buffering capacity of the synaptic vesicles in the calyx of Held terminal. We also mentioned the possible reason for 14 s delay as the reviewers pointed out. All of these points were added to the Discussion part by adding: “In some studies, even faster re-acidification time courses were suggested (Gandhi and Stevens, 2003; Atluri and Ryan, 2006). […] It may be possible that Syt2 remains stranded in the plasma membrane for ~15 s and then a partial contribution of larger endosome-like structure is needed (bulk endocytosis) to explain the 38.7 s decay time constant (Figure 1).”

We also added the following sentence to the Introduction: “Upon stimulation, we found a 12-14 s delay followed by a 20-40 s decay time constant for re-acidification of retrieved Syt2-containing organelle”.